# Modelling Genetic Benefits and Financial Costs of Integrating Biobanking into the Captive Management of Koalas

**DOI:** 10.3390/ani12080990

**Published:** 2022-04-12

**Authors:** Lachlan G. Howell, Stephen D. Johnston, Justine K. O’Brien, Richard Frankham, John C. Rodger, Shelby A. Ryan, Chad T. Beranek, John Clulow, Donald S. Hudson, Ryan R. Witt

**Affiliations:** 1Centre for Integrative Ecology, School of Life and Environmental Sciences, Deakin University Geelong, Melbourne Burwood Campus, 221 Burwood Highway, Burwood, VIC 3125, Australia; 2School of Environmental and Life Sciences, Biology Building, University of Newcastle, University Drive, Callaghan, NSW 2308, Australia; john.rodger@newcastle.edu.au (J.C.R.); shelby.ryan@uon.edu.au (S.A.R.); chad.beranek@newcastle.edu.au (C.T.B.); john.clulow@newcastle.edu.au (J.C.); 3FAUNA Research Alliance, P.O. Box 5092, Kahibah, NSW 2290, Australia; 4School of Agriculture and Food Sciences, The University of Queensland, Gatton, QLD 4343, Australia; s.johnston1@uq.edu.au; 5Taronga Institute of Science and Learning, Taronga Conservation Society, Bradleys Head Rd., Mosman, NSW 2088, Australia; jobrien@zoo.nsw.gov.au; 6School of Natural Sciences, Macquarie University, Sydney, NSW 2109, Australia; richard.frankham@mq.edu.au; 7Port Stephens Koala & Wildlife Preservation Society LTD., t/a Port Stephens Koala Hospital, One Mile, NSW 2316, Australia; don.hudson@pskh.com.au

**Keywords:** assisted reproductive technologies, artificial reproductive technologies, biobanking, captive breeding, genetic diversity, genome resource banking, heterozygosity, inbreeding, wildlife hospitals

## Abstract

**Simple Summary:**

Managed wildlife breeding faces high costs and genetic diversity challenges associated with caring for small populations. Biobanking (freezing of sex cells and tissues for use in assisted breeding) and associated reproductive technologies could help alleviate these issues in koala captive management by enhancing retention of genetic diversity in captive-bred animals and lowering program costs through reductions in the size of the required live captive colonies. Australia’s zoos and wildlife hospitals provide rare opportunities to refine and cost-effectively integrate these tools into conservation outcomes for koalas due to extensive already-existing infrastructure, technical expertise, and captive animals.

**Abstract:**

Zoo and wildlife hospital networks are set to become a vital component of Australia’s contemporary efforts to conserve the iconic and imperiled koala (*Phascolarctos cinereus*). Managed breeding programs held across zoo-based networks typically face high economic costs and can be at risk of adverse genetic effects typical of unavoidably small captive colonies. Emerging evidence suggests that biobanking and associated assisted reproductive technologies could address these economic and genetic challenges. We present a modelled scenario, supported by detailed costings, where these technologies are optimized and could be integrated into conservation breeding programs of koalas across the established zoo and wildlife hospital network. Genetic and economic modelling comparing closed captive koala populations suggest that supplementing them with cryopreserved founder sperm using artificial insemination or intracytoplasmic sperm injection could substantially reduce inbreeding, lower the required colony sizes of conservation breeding programs, and greatly reduce program costs. Ambitious genetic retention targets (maintaining 90%, 95% and 99% of source population heterozygosity for 100 years) could be possible within realistic cost frameworks, with output koalas suited for wild release. Integrating biobanking into the zoo and wildlife hospital network presents a cost-effective and financially feasible model for the uptake of these tools due to the technical and research expertise, captive koala colonies, and *ex situ* facilities that already exist across these networks.

## 1. Introduction

The continued decline of species in the wild typically drives an increased reliance on captive management, particularly on breed-for-release programs [1]. Despite their utility and place in the conservation toolbox, captive programs are not without challenges, two of which can undermine their conservation value: high costs [1,2,3,4] and retention of wild source genetic diversity [5,6,7]. The costs are substantial, with average estimates of A$200,000 per year in Australian programs and up to A$1.2 million per year for some iconic Australian species [1]. Costs are often required for multiple years or decades and will limit the number of individuals which can be held in captive management [1,2,3,4]. Cost-driven limits on colony size in turn can drive genetic diversity issues typical of small captive colonies, including adaptation to captivity [7], inbreeding [5], elevated offspring mortality [8], and reductions in reproductive fitness [9,10]. These genetic diversity challenges can combine to lower the translocation value of captive bred animals [6]. 

Biobanking and assisted reproductive technologies have been advocated as potential technological solutions to address challenges of high costs and genetic diversity in captive programs. We define biobanking as a term encompassing methods for the frozen storage of sex cells and tissues (sperm, eggs, and embryos [11,12,13]), and define assisted reproductive technologies as actions which make practical use of these frozen samples (e.g., in vitro fertilization, artificial insemination, and intracytoplasmic sperm injection [12,14,15,16,17]). Recent modelling in amphibians and marsupials highlights the potential cost reductions and genetic benefits of optimizing and integrating biobanking and assisted reproductive technologies into captive management programs [3,4,18]. Modelling suggested that integrating these technologies once optimized could dramatically slow the rate of inbreeding and reduce the size of the live colony required in captivity. This in turn substantially lowers likely program costs against conventional captive management [3,4,18]. 

These models also revealed the potential to meet or exceed long-standing and challenging genetic diversity targets (90% heterozygosity retention for 100 years [19]) in captive programs under realistic cost frameworks [3,4,18]. Though the crucial role of non-genetic fitness traits (e.g., social learning, microbiome) were not considered in those studies and are outside the scope of the current work. These modelling studies conclude that further technological innovation in assisted reproductive technologies for wildlife could provide vital low-cost biodiversity security, free economic resources to develop programs for a greater number of threatened species, and mitigate potential genetic challenges in captive programs (e.g., accumulated inbreeding, loss of reproductive fitness, and genetic reintroduction value [3,4,5,6,7,18]) by reintroducing wild-type genetic material by backcrossing captive colonies with frozen founder sperm [3,4,18]. 

Despite clear potential benefits of these technologies [3,4,18], there is a distinct lack of applied examples of biobanking and assisted reproductive technologies making measurable contributions to the wild recovery of threatened species due to remaining knowledge gaps in the underlying reproductive sciences. However, success is not impossible, and lessons can be learned from the ongoing recovery of the black-footed ferret (*Mustella nigripes*) in North America. In this case, a captive program was developed with the last remaining 18 wild ferrets which over time experienced reproductive complications due to inbreeding [20,21,22,23,24]. Applied research effort (US $4.2 million [18]) delivered an optimized protocol to supplement the captive population with frozen founder sperm using artificial insemination to add back lost heterozygosity [21]. The program, supported by a combination of natural breeding and assisted reproduction, coordinated with conventional *in situ* conservation efforts (e.g., habitat restoration and pest control), has resulted in the release of thousands of ferrets back to wild habitats [20,22]. It is significant that in addition to conventional assisted breeding, the black-footed ferret captive program has also used stem cell technology to recover biobanked somatic cells to produce a viable individual with a unique genetic profile that was not available within the extant population and could not have been produced using frozen sperm [25].

The koala (*Phascolarctos cinereus*), the world’s most iconic marsupial [26], represents a case where the integration of reproductive technologies into conservation efforts similarly stands to generate meaningful outcomes for a threatened species [16], but with more chance of sustained success if implemented before the founder population reaches such a critically small size as in the former example. The koala has become increasingly at risk of extinction across much of its range in the eastern states of Australia. Habitat loss and fragmentation [27,28], disease [29,30], vehicle strike [31], climate change [32] and bushfire [33,34,35], combine to drive significant declines in koala populations. The state of New South Wales (NSW) provides a stark example of where these threats coalesce to imperil Australia’s koala populations. The infamous 2019–2020 megafires have recently compounded these threats, with mortality in the tens of thousands of koalas [35,36]. A State parliamentary inquiry into koala populations and habitat in NSW following these fires, found that koalas face extinction in NSW, a former stronghold, by the year 2050 without intervention [35]. Due to these declines, new breeding programs are being developed specifically to supplement wild populations affected by the 2019–2020 megafires (https://amp.abc.net.au/article/12941370; accessed on 17 March 2022).

Although conventional strategies are allowing for the maintenance of optimal genetic diversity in existing koala captive populations [37], the limitations of captive breeding described above will likely affect efforts for the koala. Captive management for koalas is costly due to the unique logistical factors of the species’ specialist diet [38,39,40], therefore, strategies for future conservation breed-for-release efforts should assess and incorporate financial efficiencies where possible. In addition, the impact of sustained captivity on koalas is largely unknown and may ultimately compromise the value and fitness of captive-bred koalas for release to the wild. Koalas represent a case where assisted reproductive technologies could optimize breeding management (e.g., to retain wild-type genetic variation and mitigate disease), and where many of the foundations exist for the required technological innovation. Much of the research capital already exists to optimize assisted reproductive technologies in koalas and address remaining knowledge gaps [16]. Infrastructure and captive colonies are being established through wildlife hospital and zoo networks. Large sums of community and government investment exist for koala conservation (e.g., A$44 million throughout 2018–2021 under the NSW Koala Strategy [41]; A$8 million in crowd-funded donations to koalas following the recent bushfires, https://amp.abc.net.au/article/12941370, 17 March 2022; A$193 million in NSW Government funding across five years from 2022, https://www.environment.nsw.gov.au/news/budget-bonanza-for-the-states-biodiversity, 17 March 2022; and A$74 million in Australian Federal Government investment since 2019, https://www.abc.net.au/news/2022-01-29/koala-recovery-federal-government-funding-following-bushfires/100789364; accessed on 17 March 2022). Furthermore, historic advances have been made through the successful use of koalas as a research model for assisted reproduction [16] (e.g., the production of 34 live koala pouch young using artificial insemination with fresh and chilled sperm [14,15]). All that is needed now is a greater focus by researchers and funding bodies on optimizing the available and potential assisted reproductive technologies for the koala. There are still various remaining research challenges to overcome in koala assisted reproduction, discussed later in this study, and reviewed in [16].

This study aims to explore a reality where these technologies are optimized and could be integrated into koala captive management. We examine the potential of using biobanked koala sperm and assisted reproductive technologies to enhance the cost-effectiveness and genetic benefits possible if these technologies were to become a practical reality for koala captive management. We model the potential benefits of integrating biobanking and assisted reproductive technologies (e.g., artificial insemination and intracytoplasmic sperm injection) into the captive management of koalas by adapting the available modelling systems described above [3,4] to the unique life history traits and substantial data available on the economic cost requirements of koala captive management. We compare the costs of achieving varying genetic retention targets in managed koala conservation breeding programs, including the globally recognized benchmark of 90% of source population heterozygosity retention [19] with and without the use of biobanking and assisted reproductive technologies. We provide the most comprehensive iterations of this modelling system to date by providing detailed data on the costs of establishing a program centered around assisted reproduction, serving as a valuable blueprint for captive institutions and policymakers. Lastly, we highlight the extensive network of organizations (wildlife hospitals and rehabilitation centers) which could act as nodes and sites for the collection of koala founder sperm from wild populations, or that would benefit from the integration of assisted reproductive technologies for genetic management with low additional investment. Leveraging the modelled cost-effectiveness and potential genetic benefits of biobanking and assisted reproductive technologies showcased in this study could build the case for the optimization and integration of these tools across the network in conjunction with genetically strategic biobanking from priority and increasingly threatened wild koala populations. 

## 2. Materials and Methods

### 2.1. Case Study Species 

The koala (*P. cinereus*) is currently listed as ‘vulnerable’ under the International Union for the Conservation of Nature (IUCN) Red List [42] and as of the 12th of February 2022 is considered ‘endangered’ under Australia’s Federal Environment Protection and Biodiversity Conservation (EPBC) Act 1999 in Queensland, New South Wales, and the Australian Capital Territory (http://www.environment.gov.au/cgi-bin/sprat/public/spratlookupspecies.pl?name=Phascolarctos+cinereus&searchtype=Sciname; accessed on 17 March 2022). We selected the koala as a case study due to the availability of primary data required for modelling; particularly detailed captive holding costs (Appendix A), and data on reproductive and life history traits, for example optimal breeding age and generation length (~7 years in female koalas [43]). The koala also represents a rare case study where an extensive network of zoos and wildlife hospitals exist, as well as where comprehensive data regarding the costs required to develop a koala breeding program integrating biobanking and assisted reproductive technologies (Appendix A) already exists, providing a unique expansion of published modelling [3,4,18]. The koala has also proven to be an ideal experimental reproductive model given its adaptability and tolerance of captivity and its unique reproductive physiology [44], all of which have led to the successful development of reproductive management programs, some of which have included the use of reproductive technology [16,45]. Typically, the koala is managed in an open captive breeding system with koalas migrating to and from the system, including incorporation of a small number of additional founders recruited from the wild due to their designation as non-releasable following hospital admission. In the present study we use the program costs to model a closed captive breeding colony from the same group of founders.

### 2.2. Modelling and Analyses 

We have adapted the modelling presented in [3,4] for amphibians and for other Australian marsupials [18] to examine the genetic benefits and cost implications of integrating biobanking and assisted reproductive technologies into koala captive management. 

We have modelled 100-year captive breeding scenarios where live colony sizes of koalas reflect the potential numbers required to maintain different proportions of source population heterozygosity (*H_t_/H_o_*) with or without the use of biobanking and assisted reproductive technologies. We modelled a range of 100-year heterozygosity targets, including 90% of source population heterozygosity, which was first proposed in [19,46], and recently tested and modelled in [3,4]. We also model 95% and 99% retention, first proposed in [3,4]. We have modelled census colony sizes (*N*) and program costs required to meet the described genetic retention targets under: (1) closed captive breeding conditions (non-backcrossed populations representing the minimum number of individuals required to meet the targets without genetic intervention); and (2) in hypothetical captive colonies where assisted reproductive technologies (artificial insemination and intracytoplasmic sperm injection) are used to backcross the colony to founder male koalas using biobanked sperm every generation (7-year intervals), up to 100 years (14 generations) after the program is established. 

Colony sizes to meet the different genetic targets in conventional non-backcrossed programs were derived using an *N_e_*/*N* value of 0.3 (representing the mean value for captive vertebrates in the absence of estimates for the koala [3,4,47]). In backcrossed populations, *N_e_* was generated by random substitution into iterative genetic models until 100-year heterozygosity values matched the desired retention targets. We substituted derived *N* values for each target into economic costing models based on detailed captive holding costs for individual koalas (Appendix A). Economic models estimate the present value of 100-year program costs (assuming real discount rates of 4%) for each derived colony size with or without biobanking and assisted reproductive technologies. 

### 2.3. Cost Modelling

Year 1 economic *ex situ* holding cost per koala (*C*) is given by Equation (1) below.
*C* = *H* + *I*.(1)
where all costs are in Australian dollars (A$) and where *H* represents the economic holding cost per individual koala (given in Appendix A and encompassing costs including husbandry and veterinary staffing, eucalypt plantation management), and *I* represents variable on-site infrastructure and facilities costs. 

Year 1 program costs (*P_c_*) for any non-backcrossed conventional captive koala population and for populations integrating biobanking and assisted reproductive technologies (*P_bc_*) are given by Equations (2) and (3) below.
*P_c_* = *C* ∗ *N*. (2)
*P_bc_* = (*C* ∗ *N*) + *B*. (3)
where *C* is Year 1 economic *ex situ* holding cost per koala, as given by Equation (1), *B* represents biobanking costs (additional comprehensive fixed costs to a conventional captive program to develop a program integrating biobanking and assisted reproductive technologies, including all equipment and labor costs to perform all procedures; described in Appendix A), and *N* represents the number of adult koalas in each population. 

These equations were incorporated into iterative processes in Microsoft^®^ Excel 16.45 to model 100 years of captive management incorporating appropriate replacement costs (described in Appendix A) and assuming a constant real discount rate of 4%. The costs modelled in this study assume koalas are accessible in already operating programs with an intake of wild animals. 

### 2.4. Genetic Modelling 

We provide an updated summary of the genetic modelling system presented in [3,4]. We briefly describe the modelling system and the equations used here. For the first detailed iteration of the genetic backcross modelling see [4]. The genetic modelling involves various general assumptions around captive colony characteristics and genetic principles (Appendix A) which were first presented in [4]. We also provide detailed species-specific assumptions for the iteration of the genetic modelling presented in this study in Appendix A, including optimal breeding age, generation length and backcross frequencies for koalas which influence census colony sizes in the models. 

The predicted rate of loss of heterozygosity from koala populations was derived from the relationship between the inbreeding coefficient (*F_t_*) and heterozygosity (*H_t_*/*H*_0_) in Equation (4) [48]. This allowed the determination of residual heterozygosity (*H_t_*/*H*_0_) at each generation (*t*) for which *F_t_* was estimated. *F*_0_ for founders was assumed to be 0.
*H_t_*/*H*_0_ = 1 − *F_t_*.(4)

The increase in inbreeding between generations in captive koala colonies without backcrossing was determined from the relationship in Equation (5) [46] (p. 271), which determines *F_t_* (inbreeding coefficient, generation *t*) from *N_e_* (effective population size) and *F_t−_*_1_ (inbreeding coefficient, generation *t* − 1). This allows sequential determination of *F_t_* for any generation (*t*) up to the number required to reach 100-year heterozygosity benchmarks (14 generations for koalas assuming 7-year generational intervals).
1 − *F_t_* = (1 − 1/[2*N_e_*]) ∗ (1 − *F_t−_*_1_).(5)

We employed an iterative process [3,4], using Equation (6) to determine the effect of recurrent backcrossing (each generation) using frozen founder sperm on the rate of inbreeding for the hypothetical populations modelled in each of the three backcross scenarios.
*F_t_* = [1 − (1/2)^*t*^]/(2*N_e_*).(6)
where *N_e_* represents the effective number of male koala founders and *t* is the number of backcross generations. For detailed derivation of Equation (6) see [4]. 

### 2.5. Mapping Costs of Biobanking and Research Nodes across a Zoo and Wildlife Hospital Network

We provide detailed costings across a mapped network of zoos as well as wildlife hospital and rehabilitation centers which could provide nodes for the collection of founder sperm from wild koala populations or act as source populations for assisted reproductive research. The approach taken to construct this network is provided below. 

We conducted a comprehensive online search to select zoos and wildlife hospitals (hereafter referred to as “nodes”) in New South Wales, Queensland, Victoria, and South Australia. Selection criteria for inclusion of nodes in the network were: (1) specialize in capture, care and/or treatment of koalas (allowing us to assume that koalas would be incoming to the node and thus providing founders to sample); and (2) sufficient on-site infrastructure to hold koalas and facilitate collection and temporary or long-term storage of sperm in additional frozen storage infrastructure that could be held on-site or that may benefit from the integration of assisted reproduction for genetic management (e.g., declining populations in New South Wales or inbred South Australian koala populations [49]). Organizations were excluded if they did not meet these criteria (e.g., some excluded organizations had one small head-office which coordinated at-home care of koalas in the personal residences of volunteers or staff). We estimated the Year 1 set-up costs and on-going costs to equip nodes for biobanking by isolating the costs specific to sperm collection and cryopreservation (Appendix A) from the detailed biobanking costs (*B*) described above and presented in Appendix A. The research only nodes currently represent the costs for research focused on the collection and cryopreservation of sperm. Additional research aims such as the development of backcross tools (e.g., artificial insemination, intracytoplasmic sperm injection) would incur further research costs in line with those presented in Appendix A. The map was generated in ArcGIS Pro 2.9.0 (Esri Inc., Redlands, CA, USA) using the GPS locations (Appendix A) of each of the selected nodes revealed in the online search. 

## 3. Results

### 3.1. Modelling Costs and Genetics in Koalas 

Meeting increasingly ambitious genetic diversity targets in captive colonies of koalas would require increasing colony census numbers and total program costs (Figure 1 and Table 1 and Table 2). Without genetic intervention using assisted reproductive technologies, retaining 90% of source population heterozygosity (*N_e_* = 67; *N* = 223) in captive koalas would require significant Year-1 start-up costs of >A$5.2 million, followed by >A$73 million in 100-year costs to maintain residual heterozygosity of 0.9004 (Figure 1 and Table 1 and Table 2). More ambitious targets of retaining 95% (*N_e_* = 137; *N* = 457; *H_t_/H_o_* = 0.9501) and 99% (*N_e_* = 697; *N* = 2323; *H_t_/H_o_* = 0.9900) of source population heterozygosity without assisted reproductive technologies would require further increased Year 1 start-up costs of >A$10.8 million and >A$54.9 million respectively, as well as increased total 100-year program costs of >A$149.4 million and >$A760.2 million respectively (Figure 1 and Table 1 and Table 2).

Additional start-up and on-going investment (A$5.1 million) to use assisted reproductive technologies such as artificial insemination or intracytoplasmic sperm injection to backcross live captive female koalas to founder males using frozen sperm would allow substantial reductions in required colony census numbers and total program costs for all genetic retention targets across 100 years of captive management (Table 1 and Table 2). Backcrossing every generation at 7-year intervals into koala colonies designed to retain 90% source population heterozygosity (*N* = 17) requires >5-fold reduced start-up costs of A$904k, and >6-fold reduced total 100-year program costs of A$10.6 million (Table 2). Cost reductions are driven by substantial reductions in required colony size (Figure 1 and Table 1 and Table 2). The same proportionate results are seen for programs integrating assisted reproductive technologies to retain 95% and 99% of source population heterozygosity in captive koala colonies (Figure 1 and Table 1 and Table 2). Despite substantially lower colony sizes, reduced by over 400 and 2000 individuals against closed colony programs, backcrossed colonies still meet the same residual heterozygosity targets of 0.9500 and 0.9900 with substantially lower costs (Figure 1 and Table 1 and Table 2). In the case of the 95% (*N* = 33) retention target, start-up costs and total 100-year program costs are reduced >8-fold and >9-fold to A$1.2 million and A$16 million respectively (Table 2). Start-up costs and total 100-year program costs are both reduced >12-fold to A$4.4 million and A$59.6 million respectively in the case of the 99% (*N* = 167) retention target (Table 2).

### 3.2. Mapping Costs of Biobanking and Research Nodes across A Zoo and Wildlife Hospital Network

We propose that at least sixteen organizations and facilities could act as nodes for the collection of koala founder sperm or integrate assisted reproductive applied research programs. This could provide an extensive network across New South Wales (*n* = 5), Queensland (*n* = 7), Victoria (*n* = 2), and South Australia (*n* = 2) for management of unique and independent koala regions as informed by understanding of population and genetic differentiation across jurisdictions (Figure 2). Modest additional Year 1 investment (>A$187k) could equip these nodes to collect and cryopreserve koala founder sperm for long-term storage or thawing for practical use. We provide the total estimated costs here, for detailed itemized Year 1 costs required to equip each node see Appendix A. Modest on-going investment in labor (>$A47k) could allow the continual collection and management of samples at each node across the network (Figure 2). This equates to total Year 1 costs of A$3 million to equip the entire network for koala founder sperm collection (Figure 2). This one-off investment would be followed by a total of >A$757k in labor costs annually to ensure on-going and active collection of founder sperm and management of biobanked koala samples across the network going forward (Figure 2). 

## 4. Discussion

There are clear benefits to be gained from the optimization and integration of biobanking and assisted reproductive technologies into koala conservation breeding program management to address challenges of high costs and genetic diversity. The high economic costs and genetic diversity issues of conventional captive management (i.e., inbreeding, loss of wild-type genetic diversity and fitness [5,7,8,9]) could be addressed by backcrossing live captive koala colonies to founder males using frozen koala sperm through backcross tools such as artificial insemination and intracytoplasmic sperm injection once optimized (Figure 1 and Table 1 and Table 2). Our genetic modelling suggests integrating biobanking and assisted reproductive technologies could allow substantial reductions in inbreeding rates, allowing genetic diversity targets (90%, 95% and 99% source population heterozygosity retention [3,4,18,19]) to be met with considerably lower live census colony sizes compared to closed system captive management (Figure 1 and Table 1). Our cost modelling suggests these reductions in colony size drive 5 to 12-fold reductions in overall program costs against closed population breeding programs for the genetic diversity targets modelled here (Table 2). Our modelling assumes that conception rates from backcross events are 100%. This is a limitation of the current iteration of our models. A backcross conception rate below 100% would result in a greater number of backcross attempts being required in the same koalas and could drive slight increases in backcross specific investment. 

We recognize that koalas are held in captivity for a number of other reasons (e.g., education and advocacy as display animals) and the cost savings presented here apply to the specific case study of a closed breed-for-release program with or without integration of assisted reproduction. If these technologies were a practical reality, an already established network of 16 wildlife hospitals and zoos could act as vital collection points for koala founder sperm, with some institutions having appropriate infrastructure and staff support for temporary or permanent storage and use of samples after thawing (Figure 2). With additional investment (A$3 million setup, >A$757k/yr ongoing; Figure 2) these nodes could facilitate research programs that sample from and manage genetically distinct, inbred, or geographically isolated wild populations which could have diverse practical uses to aid koala conservation efforts.

We have presented an assessment of the existing infrastructure across a potential National node network for the integration of biobanking and/or assisted reproductive technologies to support koala conservation (Figure 2). This study does not aim to solve or propose how an optimized network of koala biobanking infrastructure should be managed, but rather highlights the capacity for a national approach. Should these technologies become available for the management of koala populations, it will be essential to develop strong governance structures to facilitate a standardized national approach to integrate these technologies into koala conservation. This would require: (1) standardized approaches and protocols for performing any assisted reproductive technique in a koala; (2) pathway development for the collection and storage of koala reproductive materials at a series of centralized locations including the integration of regional node infrastructure for collection, storage and/or research; (3) the integration of knowledge from government, wildlife carers, wildlife veterinarians, zoo practitioners, reproductive scientists and Indigenous stakeholders to inform best-practice collection and movement of sex cells between koala populations and across jurisdictions; and, (4) funding mechanism development for financial sustainability of the proposed program in perpetuity.

The influence of genetic issues and on-going loss of genetic diversity in wild koala populations suggest a need for novel tools to maximize the adaptability and resilience of the koala to prevent further decline and ultimately extinction [46,50]. Habitat fragmentation leaves koala populations geographically isolated and vulnerable to various genetic issues typical of small and/or isolated populations for which viable and economically sustainable mitigation strategies do not yet exist. For example, the genetic diversity of small, fragmented koala populations could be compromised by inbreeding and associated inbreeding depression [5], restricted gene flow [51] and genetic drift [49]. Genetic issues can lead to reproductive dysfunction and loss of reproductive fitness (e.g., testicular abnormalities in koalas due to genetic drift in small populations which reduce the ability of koalas to successfully reproduce [49,52]). These issues may also compromise survivorship [53], disease resistance [54,55] and adaptation to changing conditions (e.g., climate change; [56,57]). 

These genetic diversity challenges and their potential to arise have driven a policy and funding focus towards the genetic management of wild koala populations. One step to address this has been significant investment in recent years, including $A1 million in research investment for the frozen storage and genomic sequencing of non-living tissue samples collected from wild koalas (https://www.smh.com.au/environment/conservation/new-genetic-tools-to-stop-koalas-other-species-going-down-the-drain-20210211-p571kh.html; accessed on 17 March 2022). This research will provide a diagnostic tool to develop a genetic baseline for extant and declining koala populations; determine rates of reproduction and disease prevalence; evaluate population size and adaptive potential; identify genetic variance in populations; and may ultimately identify priority koala populations for conservation action [43].

If the approach advocated in the present study is optimized and becomes a practical reality, it has the potential to provide a mechanism to address genetic issues in wild and captive koala populations. We advocate for the imperative and neglected development of practical tools to actually retain living reproductive koala cells and tissues (e.g., sperm, eggs, and embryos), and integrate genetic material into wild populations through reintroduction and genetic rescue [58,59]. Reductions in inbreeding and adaptation to captivity (not experienced by frozen founder males) as well as the reintroduction of lost wild-type genetic diversity in output animals generated with the approach modelled here would be suited for release to the wild (Figure 1 and Table 1; [3,4,18]). Various tools (e.g., habitat restoration and protection, capacity building in rehabilitation and release activities) remain highly valuable for koala conservation efforts [28,33], but do not yet offer a viable strategy to address genetic diversity challenges or provide insurance against sudden and permanent loss of genetic diversity from wild koala populations. For example, there is no cost-effective long-term solution to protect against stochastic events such as climate change driven heatwaves, future disease outbreaks, and bushfires which pose an on-going threat of mass koala mortality (e.g., mortality of ~5000 koalas during the 2019/2020 megafires in NSW alone) resulting in sudden and permanent genetic losses ([18]; https://theconversation.com/to-save-koalas-from-fire-we-need-to-start-putting-their-genetic-material-on-ice-128049; accessed on 17 March 2022). 

Relatively low-cost collection and storage of vital koala founder sperm derived from our proposed node network could provide a form of long-term cost-effective security against stochastic events (Figure 2). Using available conventional methods, the economic and biologic feasibility to achieve the same long-term security will prove highly challenging. Based on the significantly increased costs for long-term conventional captive programs aimed at ambitious and optimal genetic diversity retention modelled in the present study (Table 1 and Table 2) and previous studies [3,4,18], it is unlikely that conservation practitioners will be able to: (1) provide long-term captive insurance in perpetuity against mass mortality or rapid, unforeseen declines of wild koala populations due to stochastic events (e.g., bushfire, climate change [32,35,36]); (2) continue to bolster wild koala populations and match the current rate of *in situ* decline of these populations and their unique genetic material; and (3) cost-effectively hold viable colony sizes long enough to meet globally accepted genetic diversity targets (e.g., 90% heterozygosity retention for 100 years [3,4,18,19]).

The substantial network of infrastructure (Figure 2) supporting koala conservation could be maximized by integrating the technology advocated here once developed. Organizations across the mapped network could serve as collection and/or storage nodes for assisted reproduction. This could add considerable value to an integrated approach to koala conservation, at a few select specialist facilities (e.g., Port Stephens Koala Hospital and Taronga Conservation Society Wildlife Hospital), that aim to integrate frozen spermatozoa into future koala-breed-for-release programs. Koalas produced in these institutions with integration of the biobanking approach could have genetic fitness which matches or surpasses current globally accepted genetic diversity standards in captive programs (Figure 1 and Table 1). These genetic targets would otherwise be highly challenging in closed breed-for-release programs based on the costs modelled here for captive management without integration of assisted reproductive technologies (Table 2). In addition, the current iteration of the modelling presented here assumes that a constant 1:1 sex ratio is maintained via colony management where males are redundant and removed from the captive program (Appendix A). These koalas could be released to the most appropriate populations in priority regions, which would hinge on the development of optimized translocation and range expansion protocols for koalas. This also presents opportunities for genetically resilient animals to be shared across the node network (Figure 2) or throughout zoo networks internationally as display animals (this also applies to other redundant animals in the program, for example koalas that are infertile). 

In the present study we present the comprehensive costs of integrating biobanking and assisted reproductive technologies as additional equipment, expertise, and infrastructure into already existing programs or institutions under two scenarios: (1) comprehensive assisted breeding programs integrating assisted reproduction where the backcrossing to captive colonies modelled here is feasible (these detailed costs differ substantially to the basic cost inputs used in previous iterations of this modelling system [3,4,18] based on the detailed data available for the present study provided in Appendix A); and (2) low-cost collection and storage of koala founder spermatozoa (Figure 2). In both scenarios we provide the most comprehensive costings data in the literature to date, providing a vital guide or blueprint for conservation practitioners, policymakers, and funding bodies. In addition to the cost-effectiveness and genetic benefits modelled in the present study (Figure 1 and Table 1 and Table 2), we outline a range of perceived potential benefits to koala conservation efforts of optimizing and then integrating biobanking and assisted reproductive technologies across the proposed network (Figure 2), provided below: Recover and make use of genetic material from koalas that can no longer contribute to the population (e.g., koalas that are diseased, recently deceased, or long dead);Proactively capture the valuable genetic material from important or imperiled wild koala populations and produce genetically fit koalas for release to the wild;Slow the rate of inbreeding by reintroducing founder genomes through frozen-thawed spermatozoa (Figure 1 and Table 1 [3,4,18]);Allow captive institutions to meet or exceed genetic diversity targets in koala breed-for-release programs (Figure 1 and Table 1 [3,4,18,60,61]);Increase the quality of the reproductive output by ensuring surrogate females can give birth to young;Overcome geographic separation and behavioral incompatibility of desirable breeding combinations and genetic pairings [62] which could also allow those koalas admitted to hospitals, zoos or captured from the wild during opportunistic research to contribute genetic material for the management of the broader wild population [60,61];Reduce transportation and social stress of male koalas between breeding program facilities [63] and reduce the need to translocate wild koalas from population to population [16];Explore assisted reproduction as a tool for the geographic selection of surrogates to potentially eliminate translocation challenges, e.g., microbiome problems [38], disease transfer between populations [16,60,61,64].

Despite historic research effort, there is currently no optimized approach to integrate the use of fresh or frozen-thawed sperm into conservation breed-for-release programs for wild koalas, to be reviewed in [65]. This upcoming review provides a case-study analysis of the baseline reproductive technology available to marsupials and considers how this technology could be optimized under a potential multi-million-dollar research pathway to integrate biobanking and assisted reproductive technology into the koala conservation toolbox.

Although there have been significant developments and advances in the use of assisted reproductive technologies (primarily artificial insemination) in the koala, there is still much to learn regarding the reliable cryopreservation of the koala sperm cell (reviewed in [16]) and the resulting production of offspring following artificial insemination with frozen-thawed semen. The challenge of koala sperm cryopreservation is particularly fascinating given that the spermatozoon of the closely related wombat is by contrast highly tolerant of the procedure [44], highlighting the importance of understanding species specificity and the need for the funding of both basic and applied research in this area. In parallel with the development of improved koala sperm cryopreservation technology, it is comforting to note koala spermatozoa also have the capacity of survival for up to 40 days in a chilled (4 °C) state; given the estrous cycle of the koala is approximately 33 days, such physiology lends itself for the immediate application of artificial insemination with chilled semen in wild koalas. 

The use of chilled semen, while not a long-term solution, nevertheless still provides a useful model in establishing workable operational logistics in preparation for a biobank based on a frozen repository. These ideas have led to the development of a pilot study known as the live koala genome bank paradigm [16,60], which incorporates the use of captive koalas as genetic reservoirs of local threatened wild genetics or brings wild koalas together in captivity for short periods of time for the purposes of breeding and genetic exchange before release back to the wild. Combined with estrus detection and/or synchronization and laboratory techniques that improve semen quality (e.g., removal or treatment of diseases such as *Chlamydia* or the recovery of sperm from recently post-mortem koalas), live and frozen genome banks both offer an exciting new paradigm for koala conservation and genetic management. 

Conducting this applied research across multi-million-dollar programs will require conservation practitioners to leverage large research and development budgets. For example, the optimized backcross protocol for the iconic black-footed ferret required an estimated US $4.2 million in dedicated applied research funding [18]. The routine use of biobanking and assisted reproduction in various sectors (e.g., medicine [66], agriculture [67,68,69,70,71], and food security [72]) provide a strong case for the ability of large funding bases to solve major research challenges and drive technological innovation. Assisted reproduction in koalas will be no different. In fact, koalas represent a case where much of the research capital already exists, including infrastructure, considerable resources, captive colonies, and technical expertise. The genetic benefits and cost-effectiveness showcased here provide a strong case for increased applied research investment and could be used as a leverage tool. 

Biobanking and assisted reproduction are unique tools, in that they could future-proof populations through capturing the genetics of key populations and long-dead individuals of which can be re-introduced into at risk populations using optimized assisted breeding. These reproductive tools are an essential addition to the koala conservation toolbox that require significant funding support from policymakers. Biobanking and assisted reproduction are proactive tools that could offer genetic management of populations through the production of individual koalas suited for wild release, and with the frequent collection and frozen storage of koala spermatozoa would act as an insurance policy against local extinction in the event of stochastic events. For high value banked genetic resources from koalas, some risk management strategies would be recommended, such as storage of replicated samples in multiple locations as proposed across our node network (Figure 2 [73,74]). This may increase biobanking costs but not considerably [3,75]. In addition, some of the numbers proposed in the present study (e.g., 17 and 33 colony animals for 90% and 95% *H_t_/H_o_* retention; Figure 1 and Table 1 and Table 2) may be considered too small to provide adequate demographic security. Captive-assurance populations may hold considerably higher numbers of live animals to deal with catastrophic demographic events in wild populations. An *N* of around 100 may be closer to a realistic minimum. For example, Species Survival Plan Programs had an average *N* of 137 [76]. Adjusting live numbers to accommodate these scenarios falls within the range of modelling scenarios conducted for other species [3,4] and would not add substantially to costs or greatly change the proportionate benefits between banked and non-banked strategies for the koala in the present study (Table 2). 

Several conservation management strategies could be bolstered using the approach advocated in this study, including genetic management and genetic rescue. Our proposed node network (Figure 2) could provide a vital supply of reproductive material for cost-effective genetic management and genetic rescue [77,78] of isolated, island or fenced koala populations [49,52,79,80,81]. Each of these scenarios presents an opportunity for our approach to improve the population management outcome relevant to the specific koala genetic management regions (e.g., Northern—Queensland/New South Wales, Southern—Victoria/South Australia). Isolated koala populations, such as those on peninsulas, separated from other populations by roadside fences, or those to be contained by reserve exclusion fencing may experience inbreeding and require management [82]. 

The research nodes we propose for South Australia are unlikely to provide useful supplementary genetics for other populations, as these populations originated from a small number of founders and are now highly inbred and require management [50,53]. These South Australian koala populations could significantly support the development of koala assisted reproductive technologies as these animals are not endangered, are overabundant, easily accessible, routinely handled, may be subject to parallel treatments and procedures (e.g., fertility control), and could act as a lower-risk model for the initial development of various assisted reproductive protocols. Biobanking and assisted reproductive technologies may provide a practical means to support the future management of inbreeding in both fast-declining and overabundant koala populations in Australia without the need for translocating animals [50,53].

It is worth noting that biobanking and assisted reproduction are not the silver bullet, no conservation technology can or should operate standalone. The black-footed ferret example typifies how such reproductive technologies have brought the species from the brink of extinction, allowing (as of 2016) the successful wild release of 4300 ferrets across 29 sites in North America [83]. However, not all ferrets have survived, and some sites require augmentation each year with captive-born black-footed ferrets or do not have any ferrets persisting [83]. It is only through a multifaceted and holistic approach to conservation of the black-footed ferret that incorporates habitat protection, habitat restoration, private land commitments, disease mitigation, human-wildlife conflict management, and conservation technologies that permanent recovery of the wild population will be possible [20,83]. We suggest that, for the koala, these same habitat protections and efforts will be required to support the rapidly declining populations. Certainly, large conservation investments in the koala have great potential to support the recovery of the species. However, without legislative support to protect koala habitat, investment in any conservation action to support koala recovery, including our proposed additions to the koala conservation toolbox, cannot function effectively for their intended purpose—to guarantee koala survival in the wild. Unless the underlying driver of decline—the continued loss of koala habitat—is addressed by the policymaker, conservation investment and efforts that aim to benefit the recovery of the koala may ultimately be undermined.

## 5. Conclusions

We have modelled scenarios where biobanking and assisted reproductive technologies are optimized and are a practical reality for integration into koala conservation breeding programs. Our genetic and economic modelling which compares closed captive koala populations suggest that supplementing with cryopreserved founder sperm using artificial insemination or intracytoplasmic sperm injection could substantially reduce inbreeding, lower required colony sizes of conservation breeding programs, and greatly reduce program costs. We have shown that ambitious genetic retention targets (maintaining 90%, 95% and 99% of source population heterozygosity for 100 years) could be possible within realistic cost frameworks with output koalas suited for wild release. Our results provide clear evidence for policymakers and funding bodies to support basic and applied research that supports the optimization and then the integration of biobanking and assisted reproductive technologies as vital tools to support koala conservation. Biobanking and assisted reproduction, once developed for the koala, could function as cost-effective and financially feasible conservation tools for applied genetic management in koala breed-for-release programs across Australia. Importantly through the reduced colony sizes and significant reduction in the required investment to maintain source population heterozygosity under a biobanking framework, could free valuable conservation funding. These cost savings could support the generation of captive programs for other at-risk species, and the on-ground actions, such as habitat restoration efforts, and monitoring of individuals, required to ensure optimal outcomes after release from these innovative potential programs. 

## Figures and Tables

**Figure 1 animals-12-00990-f001:**
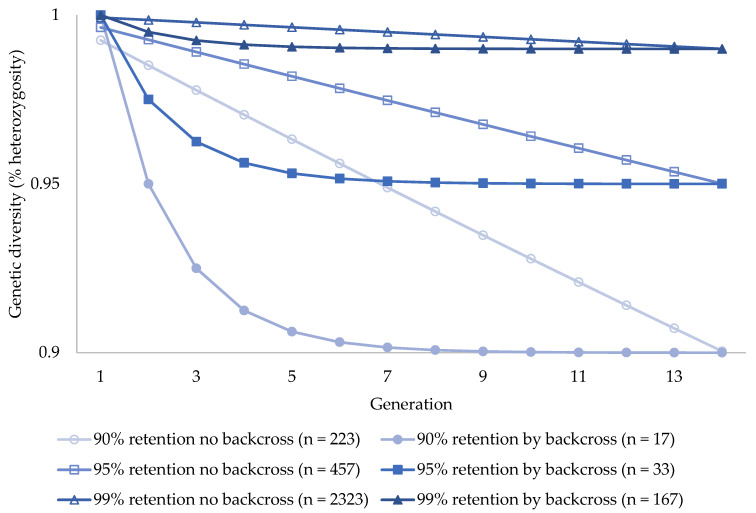
Genetic retention (initial heterozygosity *H_t_/H_o_*) across 100 years of captive management (14 generations) in hypothetical captive populations of koalas (*Phascolarctos cinereus*) designed to meet different genetic retention goals (90%, 95% and 99% of source population heterozygosity; *H_t_/H_o_*) under conventional closed captive management conditions or using assisted reproductive technologies. Genetic retention is compared in populations with no backcrossing or with backcrossing live females to founder males using cryopreserved koala founder sperm each generation (7-year intervals) of the captive period.

**Figure 2 animals-12-00990-f002:**
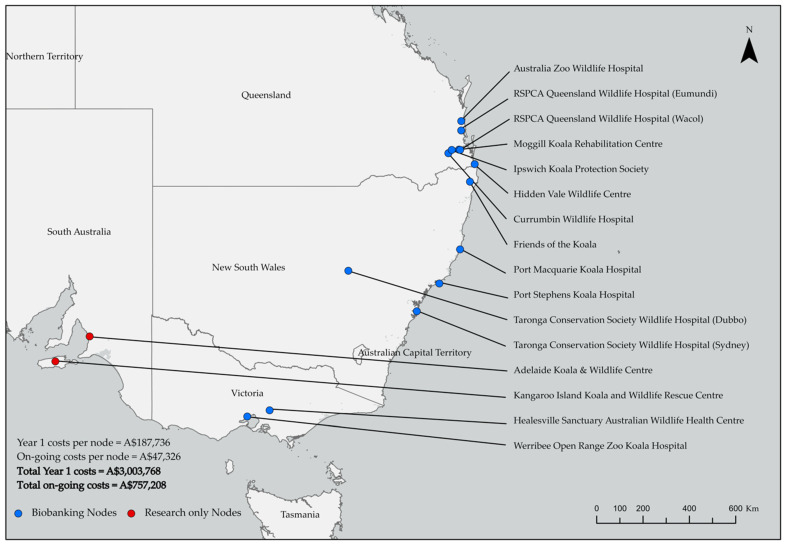
Network of organizations which could hypothetically act as nodes for the collection and/or storage of koala founder sperm and/or the integration of assisted reproductive applied research programs mapped across New South Wales, Queensland, Victoria (Blue: ‘Biobanking Nodes’), and South Australia (Red: ‘Research only Nodes’). Node locations are based on the GPS coordinates (latitude and longitude) for each organization (Appendix A). The costs ($A) are shown for Year 1 set-up (required to equip nodes to collect and cryopreserve koala founder sperm) and on-going collection and management of frozen founder sperm samples. Developed by S. A. Ryan in ArcGIS Pro 2.9.0 (Esri Inc., Redlands, CA, USA).

**Table 1 animals-12-00990-t001:** Genetic analysis for hypothetical captive colonies of koalas (*Phascolarctos cinereus*) designed to meet different genetic retention targets (90%, 95% and 99% of source population heterozygosity; *H_t_/H_o_*) under different backcross scenarios (populations with no backcross and populations with backcrossing captive females to founder males using frozen founder sperm every generation) ^1^.

Backcross Scenario	*N_e_*	*N*	*F_t_* No Backcross	*F_t_* Backcross	*H_t_/H_o_* after 100 Years
90% heterozygosity retention with no backcross	67	223	0.0996	n.d	0.9004
90% heterozygosity retention by backcrossing every generation (7-year intervals)	n.d	17	0.4999	0.1000	0.9000
95% heterozygosity retention with no backcross	137	457	0.0499	n.d	0.9501
95% heterozygosity retention by backcrossing every generation (7-year intervals)	n.d	33	0.4999	0.0500	0.9500
99% heterozygosity retention with no backcross	697	2323	0.0100	n.d	0.9900
99% heterozygosity retention by backcrossing every generation (7-year intervals)	n.d	167	0.4999	0.0100	0.9900

^1^ Effective population size (*N_e_*) and colony numbers (*N*) are shown for all hypothetical colonies. Inbreeding coefficients (*F_t_*) for backcrossed and non-backcrossed populations and heterozygosity (*H_t_/H_o_*) are values at 100 years. Colony numbers (*N*) have been derived using an assumed mean *N_e_* estimate for captive vertebrate populations of 0.3 [47]. Backcross scenarios tested: 90%, 95% and 99% heterozygosity retention with no backcross and backcross every generation. The captive colony would initially contain one live female per founder male (drawn at random from *P. cinereus* source populations). n.d = not determined.

**Table 2 animals-12-00990-t002:** Cost analysis for hypothetical captive colonies of koalas (*Phascolarctos cinereus*) designed to meet different genetic retention targets (90%, 95% and 99% of source population heterozygosity; *H_t_/H_o_*) under different backcross scenarios (populations with no backcross and populations with backcrossing captive females to founder males using frozen founder sperm every generation) ^1^.

Backcross Scenario	*N*	Cost ($) Year 1	Cost ($) Year 2	100-Year Captive Colony Costs ($)	100-Year Back-Cross Costs ($)	100-Year Program Costs ($)
90% heterozygosity retention with no backcross	223	A$5,285,556	A$2,576,923	A$73,082,167	n.d	A$73,082,167
90% heterozygosity retention by backcrossing every generation (7-year intervals)	17	A$904,505	A$350,626	A$5,453,893	A$5,158,118	A$10,612,011
95% heterozygosity retention with no backcross	457	A$10,807,778	A$5,269,231	A$149,436,671	n.d	A$149,436,671
95% heterozygosity retention by backcrossing every generation (7-year intervals)	33	A$1,298,950	A$542,934	A$10,907,786	A$5,158,118	A$16,065,904
99% heterozygosity retention with no backcross	2323	A$54,985,556	A$26,807,692	A$760,272,697	n.d	A$760,272,697
99% heterozygosity retention by backcrossing every generation (7-year intervals)	167	A$4,454,505	A$2,081,395	A$54,538,931	A$5,158,118	A$59,697,049

^1^ “Cost ($) Year 1” and “Cost ($) Year 2” are the present value of program costs in Years 1 and 2 of 100 years of colony life. “100-Year Captive Colony Costs ($)” are the net present value of total program costs under closed conventional settings without backcrossing. “100-Year Back-Cross Costs ($)” are the present value of costs of genetic backcross events (generation of koala offspring from cryopreserved founder sperm thawed for use in artificial insemination and intracytoplasmic sperm injection) across the life of the program for each backcross scenario based on number of offspring to be generated to meet retention targets. “100-Year Program Costs ($)” include the present value of total captive colony costs and expenditure for backcross events across the life of the program. Backcross scenarios tested: 90%, 95% and 99% heterozygosity retention with no backcross and backcross every generation. Colony numbers (*N*) are shown for all hypothetical colonies and have been derived using an assumed mean *N_e_* estimate for captive vertebrate populations of 0.3 [47]. All dollar amounts are shown in present value Australian currency (A$) with Year 1 starting in 2018. n.d = not determined.

## Data Availability

All data used in the analysis are provided and appear in the submitted article and Appendix A.

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
