# Peer review of "Modelling Genetic Benefits and Financial Costs of Integrating Biobanking into the Captive Management of Koalas"

_animals, 2022, doi:10.3390/ani12080990_

Round 1

Reviewer 1 Report

The authors have published in this area.

The article is lengthy. 

The survival of this important iconic species the koala is important. 

This is a scientific article. References to the popular press articles for framing are not needed. The tone of much of the article is more promotional for biobanking than objective.

The authors interrogate a future where biobanking of koala sperm, gametes and embryos is applied to captive management programs by breeding with founder males.

Biobanking, if the process resulted in more and robust offspring, would be potentially valuable. 

The genetic benefits and financial costs in this paper are considered based on various assumptions and estimates. Koala sperm from captive and wild koalas will be used.

Authors have performed modelling based on another marsupial, the wombat.

Table 2 regarding costs is likely an underestimate of the convoluted process of wild capture, handling, anesthesia, semen collection, semen processing, semen inventory, semen shipment, semen reception, monitoring of females, insemination etc

Readers will not be familiar with koalas. There is a lack of an introduction and discussion on the current challenges with ART in koala. Preserving koala semen through freezing, vitification or freeze drying has not been worked out.

There is an assumption that if semen can be cryopreserved and biobanked  it would be better for the species.

Cryopreservation is a challenge. The sperm morphology and chromatin structure of the koala is different than many species. No offspring have been produced from frozen semen. There have been no koala joey offspring produced by ovum pick up and ICSI. No IVF has been performed.

Unlike fresh or cooled semen, frozen thawed semen often has a half life of 12 hours. The seminal plasma containing OIF is removed. The fertility is lower. Post-mating inflammation may be higher. Multiple inseminations per estrus may  be required. The processing steps may or may not eliminate chlamydia and may be a source of infection for other populations. Chlamydia substantially lowers pregnancy rates and continues to pose a threat.

The biobanking will not be useful until the technology is available, this may  take decades. This may prove to be too late for the koala. Immediate action is needed.

What is clear is that there is a network of organizations and zoos that could share genetic material from male koalas now. The captive animals could be genotyped. Complementary matings could be planned to preserve biodiversity. A cooled transported semen network could be created where female reproductive activity could be tracked and semen collection / insemination planned to optimize diversity. The evidence is there that cooled semen has good pregnancy rates and results in live offspring.

This would not stop research from advancing the technology needed for ART.

Author Response

Dear Reviewer 1, 

Many thanks for the great comments you made to improve our manuscript. We provide our responses here for your further assessment as necessary. 

Kind regards,

All authors. 

Reviewer 2 Report

The investigators conducted genetic and economical modeling to compare two types of closed population management of the Koala: conventional management with no backcross versus backcrossing with founder genetic via assisted reproduction. The authors demonstrated that the latter strategy can maintain high genetic diversity at a much lower cost (owing to the need to maintain smaller numbers of animals).  Overall, this is very interesting study and will draw broad interest from wildlife conservation community. The manuscript is very well written and methodology is clearly described.  The authors did mention that the backcross model was theoretical one and will benefit when sperm cryopreservation and assisted reproductive technologies are optimized.  However, even with the availability of an optimized method, success rate will not be 100%.  Did the authors take into account of pregnancy outcome into their models?.  Without taking this into account, the authors may under estimate the need to maintain animals in the close colony, and thus, under estimating the cost.  

Author Response

Dear Reviewer 2, 

Many thanks for the great comments you made to improve our manuscript. We provide our responses here for your further assessment as necessary. 

Kind regards,

All authors. 
